# Determination of the Allelic Composition of the *sdw1/denso* (*HvGA20ox2*), *uzu1* (*HvBRI1*) and *ari-e* (*HvDep1*) Genes in Spring Barley Accessions from the VIR Collection

**DOI:** 10.3390/plants13030376

**Published:** 2024-01-27

**Authors:** Kseniia A. Lukina, Igor V. Porotnikov, Olga Yu. Antonova, Olga N. Kovaleva

**Affiliations:** Federal Research Center N.I. Vavilov All-Russian Institute of Plant Genetic Resources (VIR), St. Petersburg 190031, Russia; i.v.porotnikov@gmail.com (I.V.P.); olgaant326@mail.ru (O.Y.A.); o.kovaleva@vir.nw.ru (O.N.K.)

**Keywords:** barley, *Hordeum vulgare* L., lodging resistance, gene *sdw1*/*denso* (*HvGA20ox2*), gene *uzu1* (*HvBRI1*), gene *ari-e* (*HvDep1*)

## Abstract

The lodging of barley significantly limits its potential yield, leads to the deterioration of grain quality, and complicates mechanized harvesting. More than 30 dwarfness and semi-dwarfness genes and loci are known for barley, and their involvement in breeding can solve the problem of lodging. The most common dwarfing alleles are of the genes *sdw1*/*denso* (*HvGA20ox2*), *uzu1* (*HvBRI1*), and *ari-e* (*HvDep1*). The aim of this study was the design of dCAPS markers for the *sdw1.c* and *ari-e.GP* alleles and the molecular screening of barley accessions from the VIR collection for identifying these and other dwarfing alleles commonly used in breeding. Two dCAPS markers have been developed to identify the *sdw1.c* allele of the *HvGA20ox2* gene and *ari-e.GP* of *HvDep1*. These dCAPS markers and two known from the literature CAPS and dCAPS markers of the alleles *sdw1.a*/*sdw1.e*, *sdw1.c*, *sdw1.d*, and *uzu1.a* were used in the molecular screening of 32 height-contrasting barley accessions. This made it possible to identify the accessions with alleles *sdw1.a/sdw1.e*, *sdw1.c*, and *sdw1.d* of the *HvGA20ox2* gene, as well as accessions with a combination of *sdw1.c* and *uzu1.a* alleles of the genes *HvGA20ox2* and *HvBRI1*. A comparison of the results of genotyping and phenotyping showed that the presence of dwarfing alleles in all genotypes determines high or medium lodging resistance regardless of the influence of weather conditions. Twelve accessions were found to contain the new allele *sdw1.ins* of the *HvGA20ox2* gene, which differs from the known allele *sdw1.c* by a larger size of PCR products. It is characterized by the Thalos_2 transposon insertion; the subsequent GTTA insertion, common with the *sdw1.c* allele; and by a single-nucleotide G→A substitution at the 165 position.

## 1. Introduction

Barley (*Hordeum vulgare* L.) is an important food, feed, and brewing crop, characterized by its good adaptability to various growing conditions. However, the insufficient strength of the straw limits the potential yield of barley; due to this reason, plants often become lodged even in the years favorable for growth and development. Lodging leads to a deterioration of grain quality due to the development of diseases, causes sprouting in the ear, and also complicates mechanized harvesting and reduces the overall yield of plants. Therefore, the creation of lodging resistant cultivars is of particular importance in modern breeding programs [1].

The solution of the problem of lodging resistance in barley by breeding methods is associated, first of all, with a decrease in plant height. Semi-dwarfing genes played a crucial role in the Green Revolution in the 1960s, and their use in cereal crops significantly increased global food production [2]. More than 30 dwarfing and semi-dwarfing genes and loci have been identified for barley, including *brachytic* (*brh*), *semibrachytic* (*uzu*), *slender 1* (*sln1*), *breviaristatum* (*ari*), *erectoides* (*ert*), *semi-dwarf* (*sdw*), *slender dwarf* (*sld*), and *dense spike* (*dsp*) [3,4]. However, only a few of them are used in barley breeding due to their negative pleiotropic effect [5]. The dwarfing alleles of the genes *sdw1*/*denso* (*HvGA20ox2*), *uzu1* (*HvBRI1*), and *ari-e* (*HvDep1*) are most commonly used in breeding programs.

The *sdw1*/*denso* gene (*HvGA20ox2*) is one of the first ones cloned and characterized in barley. It is located on the long arm of chromosome 3H and is an orthologue of the *sd*-1 (*semi-dwarf*) gene in rice. The *HvGA20ox2* gene encodes gibberellin 20-oxidase, which is involved in the penultimate step of gibberellin biosynthesis [4,6]. Dwarf mutants with mutations in the *HvGA20ox2* gene are sensitive to treatment with gibberellic acid. Currently, five alleles of the *sdw1/denso* gene have been identified in cultivated barley: (1) *sdw1.c* (originally *denso*), found in the Danish cv. Abed Denso, is the only allele with a spontaneous mutation [7]. The allele has a mononucleotide deletion and a 4 bp insertion in the 5′ non-coding region of the first exon, as well as a number of other mutations in the coding region [4]. (2) *sdw1.a* allele was induced in the six-row cv. Jotun by X-ray irradiation and is characterized by a complete deletion of the *HvGA20ox2* gene [4,8]. This allele is mainly used for creating forage barley cultivars [6]. (3) The *sdw1.e* allele was identified in the cultivar Risø no. 9265, which was created in a breeding program by treating cv. Bomi with partly moderated fission neutrons. However, there are no data on the introduction of this allele into other modern cultivars. Similar to the situation with the *sdw1.a* allele, a complete deletion of the *HvGA20ox2* gene was also discovered in the case of the *sdw1.e* allele [7,9]. (4) The X-ray irradiation of cv. Valticky in 1965 resulted in obtaining cv. Diamant with the *sdw1.d* allele (*sdw1*/*denso*), which is characterized by a functional 7 bp deletion in the first exon, leading to a frameshift and a dwarf phenotype, as well as to an G→A substitution in the second intron [4]. This allele was actively used for breeding European cultivars, especially those for brewing [10]. (5) The *sdw1.Lockyer* allele in cv. Lockyer was identified by sequencing the *sdw1*/*denso* (*HvGA20ox2*) gene in different cultivars. It is characterized by a single nucleotide G→A substitution at the 70 bp position from the end of the second exon and has a similar effect on plant growth and development as the *sdw1.d* allele [11]. To identify the above-described alleles of the *sdw1*/*denso* gene the different CAPS-, both KASP and InDel markers were developed [4,6,9,11].

The *sdw1*/*denso* gene has a pleiotropic effect; that is, it affects many traits, including general adaptability, grain productivity, and grain quality [12,13]. The main characteristic is a reduction in the length of culm internodes and an increase in lodging resistance. Semi-dwarf barley cultivars with mutations in the *sdw1*/*denso* gene take part in the pedigree of most modern cultivars and are widespread in the Western United States, Canada, and Australia, as well as among European brewing cultivars [6].

The *uzu1* (*HvBRI1*) gene, a homolog of the *OsBRI1* gene in rice, is located on the long arm of chromosome 3H and encodes a protein kinase involved in the reception of brassinosteroids, the growth phytohormones [14]. The barley accessions with mutations in *HvBRI1* are insensitive to brassinosteroids treatment. At least several alleles have been described for the *HvBRI1* gene in cultivated barley: *uzu1.a*, *uzu1.b*, *uzu1.c*, *uzu1.256*, and *uzu1.297* [15]. The dwarfing allele *uzu1.a*, the most studied and represented in East Asian cultivars, carries an A→G missense mutation at the 2612 bp position in the coding region [14]. To identify the *uzu1.a* allele of the *uzu1* gene, the dCAPS and KASP markers were developed [11,16]. This allele has multiple effects on various parts of the plant, resulting in a shortened coleoptile; a small hook or projection at the coleoptile tip; shortened, wide, dark green leaves; shortened stem, spike, awns, and flower scales; and small caryopses [16,17]. Such features allow denser sowing and the application of high doses of fertilizers. Cultivars with the dwarfing allele uzu1.a occupy more than 70% of areas under barley in Japan and more than 30% of the sown area in the Korean Peninsula. Currently, over half of the naked barley forms grown in Southern Japan carry the *uzu1.a* allele [16]. Other alleles of the *HvBRI1* gene, namely *uzu1.b*, *uzu1.c*, *uzu1.256*, and *uzu1.297*, were identified in mutant barley lines [15]. At the moment, there are no reliable data on their distribution in modern barley varieties.

Another source of semi-dwarfness in spring barley among the most commonly used ones is the *ari-e.GP* allele of the *ari-e* gene located on chromosome 5H [18]. It has been shown that the *HvDep1* gene, a homolog of *Dep1* (*Dense and erect panicle 1*) in rice, which encodes the γ-subunit of the heterotrimeric G-protein involved in the regulation of organ and seed growth, is a candidate gene that determines the effect of the *ari-e* locus [19,20]. At least 10 alleles have been described for the *ari-e* gene (*HvDep1*), which have indels/nucleotide substitutions in the first and third exons: *ari-e.GP*, *ari-e.1*, *ari-e.30*, *ari-e.39*, *ari-e.119*, *ari-e.156*, *ari-e.166*, *ari-e.178*, *ari-e.222*, and *ari-e.228*. All of the mutations identified in the *HvDep1* gene of the different *ari-e* mutant lines represent loss-of-function alleles [19]. For the *HvDep1* gene, cases of polymorphism in the 5′ UTR region were also found, and the InDel markers were developed [21]. The *ari-e.GP* allele is characterized by a single A insertion at the 1508 position in the second exon of the *HvDep1* gene [20]. To identify the *ari-e.GP* allele of the *ari-e* gene, KASP markers are developed [11]. Carriers of the *ari-e.GP* allele have a semi-dwarf phenotype with short awns and exhibit salt tolerance [22]. The source of *ari-e.GP* is the Golden Promise semi-dwarf cultivar that was obtained as a gamma mutant of cv. Maythorpe in 1956 in Great Britain [23]. The cv. Golden Promise and its derivatives are still widely used for whiskey production. Other alleles of the *HvDep1* gene are used only in genetic research, but their active introduction into breeding material has been increasing in recent years.

To identify the above-described alleles of the *sdw1*/*denso*, *uzu1* and *ari-e* genes, wildly used at the breeding, KASP markers were developed [11], as well as the InDel marker for the *sdw1.c* allele of the *sdw1*/*denso* gene [4]. However, the KASP analysis is not always made available to small practice-oriented laboratories, and the identification of the *sdw1.c* allele using the InDel marker during electrophoresis in agarose gels requires long-term separation. An alternative method is the use of markers based on the cleaved amplified polymorphic sequences (CAPSs) and derived CAPSs (dCAPSs). They are successfully used for identifying the *sdw1.d* and *uzu1.a* alleles [16,24]; however, there is no information about this type of marker for identifying *sdw1.c* and *ari-e.GP*.

Unique barley accessions with a shortened culm and identified dwarfing alleles are preserved and studied in various gene banks, including the Global Collection of the N.I. Vavilov All-Russian Institute of Plant Genetic Resources (VIR). The present study was aimed at (1) developing and testing dCAPS markers for the detection of *sdw1.c* and *ari-e.GP* alleles; and (2) conducting molecular screening of the global collection of the barley accessions from the VIR collection for determining the allelic composition of the *sdw1*/*denso* (*HvGA20ox2*), *uzu1* (*HvBRI1*), and *ari-e* (*HvDep1*) genes.

## 2. Results and Discussion

### 2.1. Phenotyping

The phenotyping of accessions with respect to plant height and lodging resistance was carried out for 3 years in the fields of the Research and Practical Base “Pushkin and Pavlovsk Laboratories of VIR” (Table 1 and Appendix A). In 2021–2023, weather conditions differed between the years and affected the growth and development of barley. During the 2021 growing season, weather conditions differed significantly from the long-term average. Temperatures of the growing season were above the average long-term values, and precipitation was rare, which led to a decrease in plant height and an almost complete absence of lodging. In 2022–2023, weather conditions were close to the long-term average; however, heavy rains and strong winds occurred during the growing season and created a favorable background for identifying lodging-resistant forms.

The plant height of accessions ranged from 22 to 68 cm in 2021, from 34 to 92 cm in 2022, and from 34 to 85 cm in 2023. According to the three-year data, 24 sources of dwarfness with a straw shortened to less than 60–70 cm were identified. In 11 accessions, the plant height did not exceed 55 cm. The accessions k-30284 Namoi and k-5210 Makbo from Australia, k-28645 S-274 from Mexico, and k-28184 Local from Yemen practically did not change in height in the years with different weather conditions, thus suggesting their ecological adaptability. On the contrary, the height of accessions k-8952 and k-8953 (local from Turkey), k-30016 C.I.11077 (Peru), k-27765 Dolma (India), k-25801 Laschkes Korona, and k-25794 Rimpaus Nackta (Germany) varied greatly in different years of the study (Table 1).

According to the three-year data, the dwarf accessions k-31431 AF Cesar (Czechia), k-29730 Xima La 6 (China), k-21338 Ehimehadaka N4, k-21341 Jamatohadaka, k-21363 Shikokuhadaka N3, k-21378 Shinjinryoku N1 (Japan), k-28868 Complex hybrid (Mexico), k-31059 Tamalpais (USA), and Kibtsel (Leningrad Province) were identified as sources of lodging resistance of 7-9 points. The accessions that demonstrated moderate lodging resistance of 5-7 points were k-31540 Lotos (Germany), k-31430 AF Lucius (Czechia), k-28957 Karan 3, k-28961 Karan 19 (India), k-15950 N1060 (China), k-27080 Belorusskij 76 (Belarus), k-28645 S-274 (Mexico), k-30284 Namoi (Australia), and k-26209 Potra (Finland). The remaining accessions were non-resistant to lodging in different years (Table 1).

### 2.2. Markers Development

The sequence of the wild-type allele of the *HvGA20ox2* gene (GenBank: LOC123445118) was extracted from the whole genome assembly of the barley cv. Morex (MorexV3). To localize mutations characteristic of the *sdw1.c* and *sdw1.d* alleles, the data from the work by Y. Xu et al. (2017) were used [4]. This way, a consensus diagram of the *HvGA20ox2* gene was constructed which reflects the localization of mutations associated with the *sdw1.c* and *sdw1.d* alleles (Figure 1).

To search for the *HvDep1* gene sequence in the NCBI databases, we used the published sequence of its mRNA (GenBank: FJ039903.1). The BLAST search in the full genome assemblies of MorexV3 and Golden Promise (GPv1) resulted in our finding full-length gene sequence exons that were identical to the mRNA. The alignment of the *HvDep1* gene sequences from cvs. Morex and Golden Promise (mutant allele *ari-e.GP* carrier) confirmed the presence of an A-insertion at the 1508 position in the second exon of the *HvDep1* gene [20].

To develop a DNA marker for the *sdw1.c* allele of the *HvGA20ox2* gene, a 4 bp insertion located in the first exon at the 64 position (+GTTA) was selected. Because the region with the insertion was found to contain no restriction sites, dCAPS primers were created for the identification of this allele. The penultimate nucleotide in the reverse primer dCsdw1.c-R was modified in such a way that the PCR products of all alleles, except *sdw1.c*, formed a restriction site, CCGG, characteristic of HpaII (Figure 2).

In a similar way, a dCAPS marker was developed for the insertion 1508 (+A) in the *ari-e.GP* allele of the *HvDep1* gene. Due to a modification of the reverse primer sequence, the PCR product contained the restriction site CATCC (instead of the original CAACC), which allowed this mutation to be distinguished from all other alleles by FokI restriction (Figure 3 and Appendix A). The mechanism of the dCAPS marker for *ari-e.GP* allele action is similar: a restriction site is formed in the wild type during amplification, while the cutting of PCR products does not occur in the mutant allele *ari-e.GP*.

### 2.3. Markers Approbation

The testing of all CAPS and dCAPS markers for the detection of the *sdw1.d*, *sdw1.c*, *uzu1.a*, and *ari-e.GP* alleles was carried out on a sample of 12 barley accessions from the VIR collection with the known alleles of dwarfing genes.

To identify the *sdw1.d* allele (SNP G→A) of the *HvGA20ox2* gene, the corresponding CAPS marker was used (Appendix A). The presence of the *sdw1.d* allele was determined from the absence of PCR products’ restriction with the HvGA20ox2F/R primers, while all other alleles, including the wild type, had a site for the HaeIII restriction enzyme in the SNP region (Figure 4a). The *sdw1.d* allele was detected in control accessions k-19691 Diamant (Czechoslovakia), k-30293 Franklin (Australia), k-31247 Tipple (Great Britain), and k-31355 Braemar (Germany). Accessions with the *sdw1.a* allele had no amplification products with the HvGA20ox2F/R primers because they are characterized by a complete deletion of the gene, which was confirmed in accessions k-30291 Yerong (Australia) and k-19037 Jotun (Sweden) carrying *sdw1.a* (Figure 4a) [6,24].

To detect the *sdw1.c* allele of the *HvGA20ox2* gene, we used the dCAPS marker developed in this study, which detects a 4 bp insertion at the 64 position (+GTTA) (Appendix A). The *sdw1.c* allele was detected in the controls k-19308 Dayton (USA) and k-19808 Deba Abed (Denmark); there was no cutting of PCR products in their case (Figure 4b).

The *uzu1.a* mutant allele of the *HvBRI1* gene was determined using the dCAPS marker proposed by Saisho et al. (2004). The marker identifies the SNP A→G substitution at the 2612 position from the start codon [16]. The presence of *uzu1.a* was determined by the restriction of PCR products (BstHHI restriction enzyme) (Appendix A). The used sample contained no controls with the *uzu1.a* allele from literary sources. However, several accessions, which presumably carry this allele, were selected based on the phenotypic assessment of seedlings that have a small hook or protrusion at the coleoptile tip, as is characteristic of *uzu1.a*. Indeed, the dCAPS marker confirmed the presence of *uzu1.a* in k-21338 Ehimehadaka N4, k-21341 Jamatohadaka, k-21363 Shikokuhadaka N3, and k-21378 Shinjinryoku N1 (Figure 5).

Also, the dCAPS marker was used to differentiate the *ari-e.GP* allele of the *HvDep1* gene. We developed primers that flank the region with the A insertion at the 1508 position in the start codon (Appendix A). During amplification, a CATCC restriction site, the recognition site for FokI restriction endonuclease, is formed in wild-type alleles; no cutting of 154 bp PCR products in the mutant allele occurs (Figure 4d). Testing of the dCAPS marker for the identification of *ari-e.GP* confirmed its presence in the control accession k-20491 Golden Promise (Great Britain).

To sum it up, the testing of CAPS and dCAPS markers for the *sdw1.d*, *sdw1.c* (*HvGA20ox2* gene), *uzu1.a* (*HvBRI1*), and *ari-e.GP* (*HvDep1*) alleles using a sample of control accessions confirmed that the tested markers effectively differentiate mutant variants from the wild-type and other alleles (Figure 4 and Figure 5).

### 2.4. Molecular Screening

Molecular screening of 32 height-contrasting barley accessions from the VIR collection was carried out using CAPS and dCAPS intragenic markers (Appendix A) of alleles of the dwarfing genes *sdw1/denso* (*HvGA20ox2*), *uzu1* (*HvBRI1*), and *ari-e* (*HvDep1*).

Barley accessions with various dwarfing alleles of the *sdw1/denso* gene were most represented in the sample (Table 1). Four accessions, namely k-31430 AF Lucius (Czechia, var. *nudum*), k-27080 Belorusskij 76 (Belarus, var. *nudum*), k-28650 S-281 (Mexico, var. *himalayense*), and k-30284 Namoi (Australia, var. *nudum*), are characterized by the presence of the *sdw1.d* gene allele. Based on the absence of amplification products from primers HvGA20ox2F/R, three accessions were identified as presumably carrying the *sdw1.a* or *sdw1.e* allele. These accessions characterized by a large deletion affecting the *HvGA20ox2* gene are k-28957 Karan 3 (India, var. *coeleste*), k-28961 Karan 19 (India, var. *coeleste*), and k-31059 Tamalpais (USA, var. *coeleste*). These are forms of six-row barley from India and the USA.

The *sdw1.c* allele was identified in two accessions: k-29730 Xima La 6 (China, var. *coeleste*) and k-26209 Potra (Finland, var. *parallelum*). Additionally, this allele was detected in four more accessions, though in combination with the dwarfing allele *uzu1.a* of the *uzu1* gene; these accessions are k-21338 Ehimehadaka N4 (Japan, var. *subnudipyramidatum*), k-21341 Jamatohadaka (Japan, var. *subnudipyramidatum*), k-21363 Shikokuhadaka N3 (Japan, var. *brevisetum*), and k-21378 Shinjinryoku N1 (Japan, var. *brevisetum*). The mentioned accessions belong to six-row naked forms of Japanese origin. Indeed, the presence of the *uzu1.a* allele is typical for samples of Chinese, Japanese, and Korean origin [16]. A study by Zhang J. and Zhang W. (2003) showed that 68.4% of barley cultivars bred in China originated from only six dwarf cultivars, four of which carried a mutant allele of the *uzu1* gene, and the other two probably carried a mutation in the *sdw1/denso* gene [25].

During the screening of the sample using the dCAPS marker of the *sdw1.c* allele designed in the present study, 12 accessions were found to differ in terms of the *sdw1.c* allele by a larger size of PCR products (Figure 4b). This may indicate the presence of an insertion in the *sdw1/denso* gene sequence.

The studied sample was found to contain no barley accessions with the *ari-e.GP* allele (*ari-e* gene); a restriction of PCR products occurred in all genotypes, except for the control k-20491 cv. Golden Promise (Figure 4d).

The remaining seven barley accessions did not show the presence of the studied alleles of the dwarfing genes.

A comparison of the phenotyping and molecular screening data (Table 1 and Appendix A) showed that all barley accessions with the *sdw1.d* and *sdw1.a*/*sdw1.e* alleles of the *sdw1/denso* gene (*HvGA20ox2*) are characterized by medium resistance to lodging. High resistance to lodging was demonstrated by accessions with the *sdw1.c* allele, or with its combination with the *uzu1.a* allele of the *HvBRI1* gene. The height of these accessions did not exceed 62 cm in the years with high humidity, and 43 cm in the driest period. The accessions with these alleles or combinations of alleles are a promising source material for their introduction into the breeding process as sources of dwarfness and lodging resistance.

Among the accessions that did not show the presence of the studied alleles of the dwarfing genes, the results of three-year observations have shown three dwarf ones to be most stable, namely k-31431 AF Cesar (Czechia), k-28868 complex hybrid (Mexico), and Kibtsel (Leningrad Province). Most likely, their resistance to lodging is due to the presence of other dwarfing genes and requires further study.

### 2.5. New Allelic Variant of the *sdw1*/*denso* (HvGA20ox2) Gene

To confirm the insertion in the *sdw1*/*denso* gene, identified using dCAPS and InDel markers, we carried out the sequencing of PCR products. In the case of the MC40861P3F/R primers selected from the literature, sequencing was carried out for the following barley accessions: cv. Dolma (GenBank: PP053705), cv. Makbo (PP053704), cv. Rimpaus Nackta (PP053706), C.I.10405 (PP053707), and k-8953 (PP053708); those carrying the new identified insert; and cvs. Deba Abed (*sdw1.c* allele, PP053703) and Morex (wild type allele, PP053702). The sequencing of PCR products from the dCsdw1.c-F/R primers designed in the present study was carried out for the accession k-18374 (PP053712) with an insert; for Dayton (PP053710) and Xima La 6 (PP053711), both with the *sdw1.c* allele; and for Morex (PP053709).

Indeed, all accessions with a larger PCR product contained a 166 bp insertion (Figure 6 and Appendix A) located at the 64 bp position from the beginning of the 5′ non-coding region of the first exon of the *HvGA20ox2* gene. It was the same in all the six sequenced barley accessions. It is noteworthy that the *sdw1.c* allele has a characteristic GTTA insertion located at the position that completely coincides with the 3′ end of the 166 bp insertion identified in the present study at the same position. The BLAST search for the sequence of the 166 bp insertion identified high homology (92.02%, 98% coverage) to a region annotated as transposon Thalos_2 (GenBank: KR813336.1). This transposon is 162 bp long and does not contain the GTTA sequence at its end, characteristic of the *sdw1.c* allele. The alignment of the sequencing results also revealed a single nucleotide G→A substitution at the 135 position from the beginning of the first exon. Like the 166 bp insertion, this SNP is located in the first exon upstream of the start codon.

As a result, we identified a new allele of the *sdw1/denso* gene, characterized by an insertion of the Thalos_2 transposon, a subsequent GTTA insertion, common with the *sdw1.c* allele, as well as by a single-nucleotide G→A substitution at the 165 position, which we named *sdw1.ins*. The same sequence with exactly the same transposon insertion and single nucleotide substitution was found in the barley pan-genome [26] in three accessions (HOR8148, HOR10350, and HOR13821). However, the allelic composition of the *HvGA20ox2* gene was not described by the authors.

The allele *sdw1.ins* was detected in 12 spring barley accessions of different ecological and geographical origin (Table 1). The plant height in accessions with the insertion ranged from 36 to 91 cm and varied depending on weather conditions. Under different weather conditions, accessions with the *sdw1.ins* allele did not retain high lodging resistance. According to the data from 2022 to 2023, it was medium or low. Presumably, this insertion at the beginning of the non-coding region does not affect lodging resistance. The greater part of the accessions is represented by local forms, mainly from high mountainous regions. Their ecogeographic diversity suggests that the transposon insertion occurred independently.

## 3. Materials and Methods

### 3.1. Plant Material

The material for the study was 32 height-contrasting barley accessions from the VIR collection. They were represented by 14 botanical varieties from 19 countries. A set of 12 cultivars and almost isogenic lines, selected from literature sources, was taken as the control.

### 3.2. Phenotyping

Phenotyping with respect to plant height and lodging resistance was carried out in accordance with the “Methodological guidelines for studying and maintaining the global collection of barley and oat” [27] in 2021–2023 on the fields of the Research and Practical Base “Pushkin and Pavlovsk Laboratories of VIR”.

### 3.3. Primers Design

The sequences of different alleles of the *HvGA20ox2* and *HvDep1* genes were aligned using the MEGA XI [28] and Unipro UGENE programs [29].

The whole genome assembly of cv. Morex (MorexV3) was used as a source of wild-type alleles of *HvGA20ox2* and *HvDep1* genes. The sequence of the *sdw1.c* allele was given according to Y. Xu et al., 2017 [4]. The *ari-e.GP* allele of the *HvDep1* gene was extracted from the genome assembly of cv. Golden Promise (GPv1).

Restriction sites were determined using SnapGene software v.7.0.1 (from GSL Biotech, www.snapgene.com (accessed on 1 December 2023)).

Primers for dCAPS markers were designed using the dCAPS Finder 2.0 program [30]. Annealing specificity was tested via in silico PCR, using genome assemblies of Morex (MorexV3) and Golden Promise (GPv1) cultivars. The quality control of primers, i.e., their tendency to form secondary structures (hairpins and homo- and heterodimers) was assessed using the OligoAnalyzer resource (Integrated DNA Technologies, Inc., Coralville, IA, USA).

### 3.4. Molecular Screening

DNA was isolated from the upper leaves of individual plants (3 plants per accession) collected during the heading period in the field. The method of SDS extraction was used [31]. The concentration and quality of the isolated DNA were controlled by an Implen N60 nanophotometer (IMPLEN, Munich, Germany), as well as by the 0.8% agarose gel electrophoresis.

Intragenic DNA markers selected from literature sources were chosen for molecular screening of the *sdw1.c* and *sdw1.d* alleles of the *HvGA20ox2* gene and the *uzu1.a* allele of the *HvBRI1* gene. Additionally, dCAPS markers were developed for *sdw1.c*, as well as for the *ari-e.GP* allele of the *HvDep1* gene (Figure 1, Figure 2 and Figure 3). The primers used in the study and PCR programs for them are listed in Appendix A.

PCR was carried out under standard conditions. The PCR with primers MC40861P3F/R and primers for detecting the *uzu1.a* allele used 20 μL of a reaction mixture that contained 40 ng of template DNA, 1× reaction buffer (Dialat, Moscow, Russia), 1.5 mM MgCl_2_, 0.3 mM of each dNTP, 200 nM of each forward and reverse primers (Evrogen, Moscow, Russia), and 1 unit of Taq polymerase (Dialat, Russia). For all other PCR primers, the mixture contained 2.5 mM MgCl_2_.

The restriction of PCR products was carried out overnight in 15 μL of the reaction mixture according to the manufacturer’s protocol (SibEnzyme, Novosibirsk, Russia).

PCR products were separated via electrophoresis in 2% and 3% agarose gels at 5 V/cm of gel length for 2–3 h. The gels were stained with ethidium bromide and visualized in UV light.

### 3.5. Sequencing

Amplification products generated by the MC40861P3 F/R and dCsdw1.c-F/R primers were used for sequencing. PCR products were purified using the Cleanup Standard Kit (Evrogen, Russia). Sanger sequencing was performed on a 24-capillary 3500xL Genetic Analyzer (Applied Biosystems, Waltham, MA, USA) at the Center for Shared Use “Genomic Technologies, Proteomics and Cell Biology” of the All-Russia Research Institute for Agricultural Microbiology. The sequences were read and aligned in the programs MEGA XI [28], Unipro UGENE [29], and BioEdit Sequence Alignment Editor [32].

## 4. Conclusions

The present study resulted in the development of dCAPS markers for identifying the *sdw1.c* dwarf alleles of the *HvGA20ox2* gene and *ari-e.GP* of the *HvDep1* gene in barley. These alleles are widespread and have been used for many years in breeding aimed at reducing plant height. The advantage of dCAPS markers is the ability to identify alleles on standard PCR equipment using agarose gel electrophoresis, as well as good reproducibility and unambiguous interpretation of the obtained results.

The testing of all CAPS and dCAPS markers on a sample of barley accessions with the known dwarfing alleles *sdw1.d* and *sdw1.c* of the *HvGA20ox2* gene and *uzu1.a* of the *HvBRI1* gene, as well as *ari-e.GP* and wild-type alleles, showed complete compliance with the expected results and high efficiency of mutant variant differentiation.

The molecular screening of 32 height-contrasting barley accessions from the VIR collection has found the *sdw1.d* allele of the *sdw1*/*denso* gene in four accessions, the *sdw1.a* or *sdw1.e* allele of the *sdw1*/*denso* gene in three accessions, the *sdw1.c* allele of the *sdw1*/*denso* gene in two accessions, and the *sdw1.c* allele in combination with the *uzu1.a* allele of the *uzu1* gene in four accessions. The *ari-e.GP* allele was absent in the studied sample; it was identified only in control cultivars. A comparison of the results of genotyping and phenotyping has shown that the presence of dwarfing alleles in all genotypes determines high or medium lodging resistance, regardless of the influence of weather conditions.

The accessions with the identified alleles of the *sdw1* (*HvGA20ox2*) and *uzu1* (*HvBRI1*) genes are recommended for the use as sources of dwarfness and resistance to lodging in breeding programs.

In 12 samples, a previously undescribed allele of the *sdw1*/*denso* gene was identified which differs from the *sdw1.c* allele because of its larger size of PCR products generated by the same primers. The presence of a 166 bp insertion was confirmed by sequencing with two pairs of primers, dCsdw1.c-F/R and MC40861P3F/R. A new allele of the *HvGA20ox2* gene that was identified in this study was named *sdw1.ins*. It is characterized by a Thalos_2 transposon insertion; a subsequent GTTA insertion, which is common with the *sdw1.c* allele; and a single-nucleotide G→A substitution at the 165 position.

## Figures and Tables

**Figure 1 plants-13-00376-f001:**
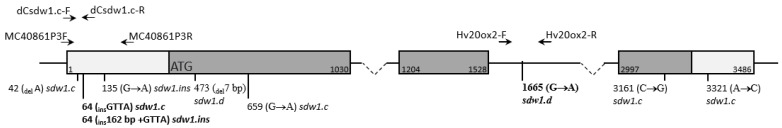
Diagram of the *HvGA20ox2* gene with the localized mutation characteristic of the *sdw1.c*, *sdw1.ins*, and *sdw1.d* alleles, as well as the location of primers for their detection. The non-coding regions of exons are highlighted in light gray; the coding ones are highlighted in dark gray.

**Figure 2 plants-13-00376-f002:**
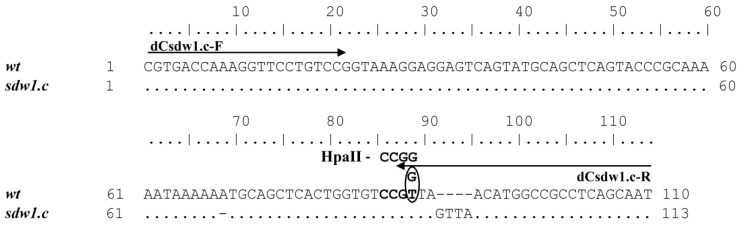
Sequence alignment flanked by the dCsdw1.c-F/R dCAPS primers for wild-type (from the MorexV3 genome assembly) and *sdw1.c* (cv. Deba Abed according to Xu et al., 2017 [4]) alleles of the *HvGA20ox2* gene. The oval indicates the nucleotide that has been changed in the dCAPS-primer compared to the original gene sequence.

**Figure 3 plants-13-00376-f003:**
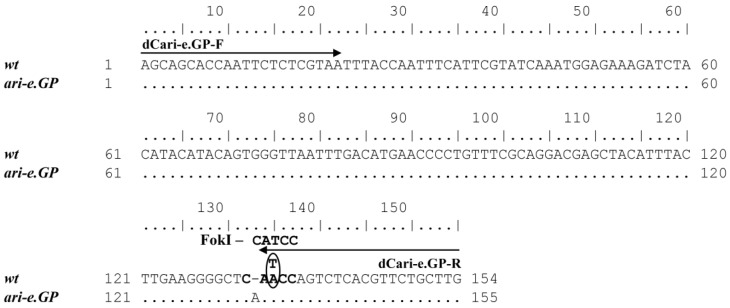
Sequence alignment flanked by the dCari-e.GP-F/R dCAPS primers for the wild-type (from the MorexV3 genome assembly) and *ari-e.GP* (from the GPv1 assembly) alleles of the *HvDep1* gene. The oval indicates the nucleotide that was changed in the dCAPS-primer compared to the original gene sequence.

**Figure 4 plants-13-00376-f004:**
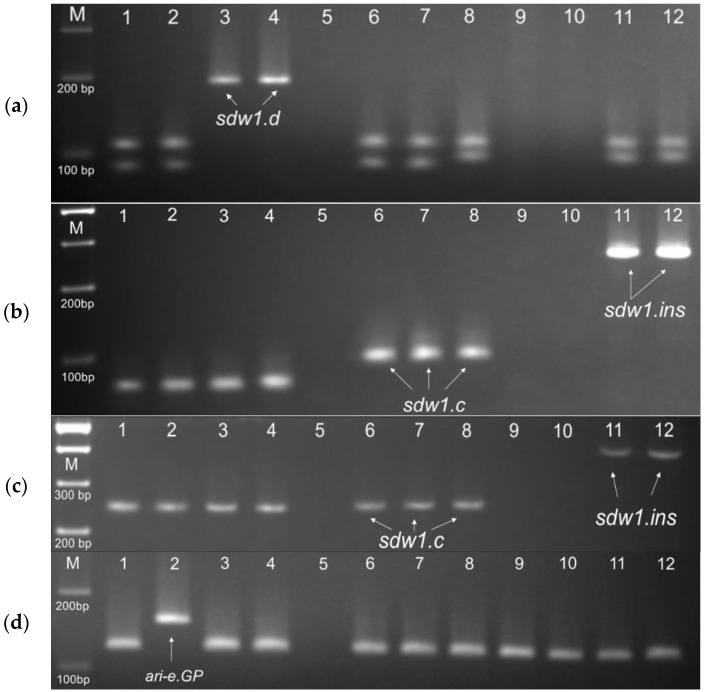
Identification of the *sdw1/denso* (*HvGA20ox2*) and *ari-e* (*HvDep1*) genes allelic composition: (**a**) *sdw1.d* allele of the *sdw1/denso* gene—CAPS marker (primers Hv20ox2-F/R + HaeIII); (**b**) *sdw1.c* allele of the *sdw1/denso* gene—dCAPS marker (dCsdw1.c-F/R + HpaII); (**c**) *sdw1.c* allele of the *sdw1/denso* (*HvGA20ox2*) gene—InDel marker (MC40861P3F/R); (**d**) *ari-e.GP* allele of the *ari-e* gene—dCAPS marker (primers dCari-e.GP-F/R + FokI). The numbers indicate barley accessions with different alleles of the *sdw1/denso* (*HvGA20ox2*), *uzu1* (*HvBRI1*), and *ari-e* (*HvDep1*) genes: 1—Morex (wild type); 2—Golden Promise (*ari-e.GP*); 3—Diamant (*sdw1.d*); 4—Namoi (*sdw1.d*); 5—negative control (H_2_O); 6—Deba Abed (*sdw1.c*); 7—Potra (*sdw1.c*); 8—Shikokuhadaka N3 (*sdw1.c* and *uzu1.a*); 9—Yerong (*sdw1.a*/*sdw1.e*); 10—Karan 3 (*sdw1.a*/*sdw1.e*); 11—k-28184 (*sdw1.ins*); 12—C.I.10405 (*sdw1.ins*). M—100 bp molecular weight marker (SibEnzyme).

**Figure 5 plants-13-00376-f005:**
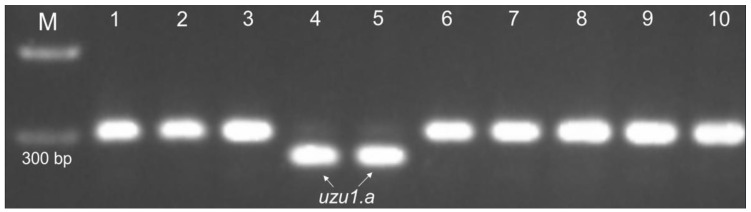
Identification of the *uzu1.a* allele (gene *HvBRI1*) with dCAPS marker (+BstHHI). The numbers indicate barley accessions with different alleles of the *sdw1/denso* (*HvGA20ox2*), *uzu1* (*HvBRI1*), and *ari-e* (*HvDep1*) genes: 1—Morex (wild type); 2—Golden Promise (*ari-e.GP*); 3—Diamant (*sdw1.d*); 4—Shikokuhadaka N3 (*sdw1.c* and *uzu1.a*); 5—Jamatohadaka (*sdw1.c* and *uzu1.a*); 6—Potra (*sdw1.c*); 7—Yerong (*sdw1.a*/*sdw1.e*); 8—Karan 3 (*sdw1.a*/*sdw1.e*); 9—k-28184 (*sdw1.ins*); 10—C.I.10405 (*sdw1.ins*). M—100 bp molecular weight marker (SibEnzyme).

**Figure 6 plants-13-00376-f006:**
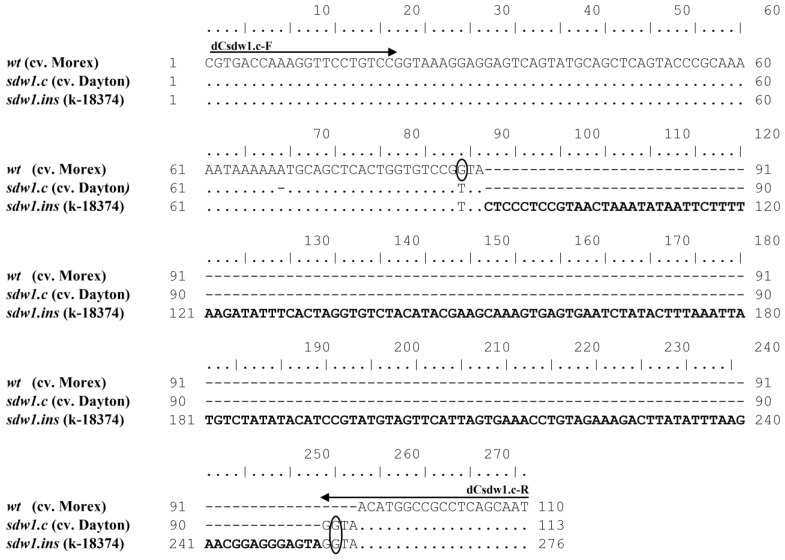
Sequence alignment of PCR-products flanked by the dCsdw1.c-F/R primers. The sequence homologous to the Thalos_2 transposon is highlighted in bold. The oval indicates the nucleotide that was changed in the dCAPS-primer compared to the original gene sequence.

**Table 1 plants-13-00376-t001:** Characteristics of barley accessions with respect to plant height and lodging resistance, Pushkin, 2021–2023.

No.	VIR Catalogue No.	Origin	Name	Plant Height, cm			Lodging Resistance, Points			Alleles/Combinations of Alleles of the Genes *sdw1/denso, uzu1* and *ari-e*	
				2021	2022	2023	2021	2022	2023		
1	22089	Leningrad Province	Belogorskij	68.0 ± 2.3	77.7 ± 3.5	81.6 ± 3.5	9	7	3–5	-	
2	-	Leningrad Province	Kibtsel	46.0 ± 1.3	69.0 ± 1.4	55.3 ± 2.4	9	7	7	-	
3	31540	Germany	Lotos	52.0 ± 0.6	47.4 ± 2.4	67.0 ± 1.0	7	9	5–7	-	
4	25794	Germany	Rimpaus Nackta	48.8 ± 2.1	68.4 ± 3.2	85.3 ± 1.3	7	7	5	*sdw1.ins*	
5	25801	Germany	Laschkes Korona	51.6 ± 0.7	91.2 ± 6.3	73.0 ± 1.0	9	5	3	*sdw1.ins*	
6	31430	Czechia	AF Lucius	45.6 ± 0.7	59.4 ± 2.1	63.3 ± 2.4	9	9	5–7	*sdw1.d*	
7	31431	Czechia	AF Cesar	43.6 ± 1.3	56.0 ± 1.3	56.7 ± 2.4	9	9	7	-	
8	27765	India	Dolma (HBLI)	36.0 ± 2.1	65.0 ± 3.1	55.3 ± 2.4	9	9	5	*sdw1.ins*	
9	28957	India	Karan 3	42.4 ± 1.7	55.2 ± 1.4	56.0 ± 1.1	9	9	5–7	*sdw1.a/sdw1.e*	
10	28961	India	Karan 19	40.8 ± 0.5	50.0 ± 1.3	51.3 ± 1.7	7	9	5–7	*sdw1.a/sdw1.e*	
11	28184	Yemen	Local	43.8 ± 1.4	46.6 ± 1.9	45.0 ± 1.2	9	9	3	*sdw1.ins*	
12	29730	China	Xima La 6	26.8 ± 1.0	34.4 ± 3.2	55.3 ± 2.9	9	9	7	sdw1.c	
13	27080	Belarus	Belorusskij 76	48.0 ± 1.1	59.8 ± 2.3	65.7 ± 2.3	7	7	5–7	*sdw1.d*	
14	21338	Japan	Ehimehadaka N4	29.2 ± 0.8	48.0 ± 1.7	50.0 ± 5.8	9	9	7	*sdw1.c* and *uzu1.a*	
15	21341	Japan	Jamatohadaka	29.4 ± 1.1	52.2 ± 2.5	46.7 ± 0.9	9	9	7	*sdw1.c* and *uzu1.a*	
16	21363	Japan	Shikokuhadaka N3	24.2 ± 1.0	40.6 ± 2.0	34.3 ± 1.2	9	9	7	*sdw1.c* and *uzu1.a*	
17	21378	Japan	Shinjinryoku N1	22.4 ± 1.4	41.6 ± 0.9	35.3 ± 1.8	9	9	9	*sdw1.c* and *uzu1.a*	
18	23378	Bolivia	C.I.10405	51.2 ± 1.0	70.0 ± 2.3	71.0 ± 2.1	7	5–7	3–5	*sdw1.ins*	
19	28645	Mexico	S-274	49.2 ± 1.0	52.2 ± 1.8	45.0 ± 4.1	9	9	5–7	-	
20	28650	Mexico	S-281	38.4 ± 1.0	56.8 ± 2.0	49.6 ± 2.0	7	9	5	*sdw1.d*	
21	28868	Mexico	Complex hybrid	40.4 ± 1.7	51.6 ± 1.6	49.0 ± 0.8	9	9	7	-	
22	30016	Peru	C.I.11077	42.6 ± 2.6	76.6 ± 1.9	83.3 ± 0.9	9	5–7	3	*sdw1.ins*	
23	31059	USA	Tamalpais	40.4 ± 1.7	46.8 ± 1.7	50.0 ± 1.1	9	9	7	*sdw1.a/sdw1.e*	
24	31518	USA	Washonupana	42.4 ± 1.9	54.6 ± 0.6	54.0 ± 1.2	9	5–7	3–5	-	
25	5210	Australia	Makbo	64.8 ± 1.9	71.6 ± 2.2	65.7 ± 3.8	5	3–5	5–7	*sdw1.ins*	
26	30284	Australia	Namoi	65.2 ± 3.2	61.6 ± 2.9	64.3 ± 0.7	9	9	5–7	*sdw1.d*	
27	18374	Dagestan		49.6 ± 2.6	55.0 ± 2.3	74.3 ± 3.0	9	1	3	*sdw1.ins*	
28	31068	Pakistan	IG 24534	49.2 ± 0.5	52.2 ± 4.8	64.0 ± 2.3	5	7	3	*sdw1.ins*	
29	31069	Pakistan	IG 24550	37.2 ± 1.5	37.8 ± 2.1	55.0 ± 1.0	9	7	3	*sdw1.ins*	
30	8952	Turkey		46.4 ± 1.2	77.0 ± 1.5	82.5 ± 2.0	9	5	3–5	*sdw1.ins*	
31	8953	Turkey		46.8 ± 1.0	77.0 ± 2.3	75.0 ± 1.1	7	5	3–5	*sdw1.ins*	
32	26209	Finland	Potra	43.6 ± 0.7	62.6 ± 2.5	62.6 ± 1.4	7	9	5–7	*sdw1.c*	
**Controls**											
**No.**	**VIR Catalogue No.**	**Origin**		**Name**			**Gene**			**Allele**	**Literature Source**
1	30291	Australia		Yerong			*HvGA20ox2*			*sdw1.a*	[4]
2	-	Sweden		*sdw 1.a*			*HvGA20ox2*			*sdw1.a*	[6]
3	19308	USA		Dayton			*HvGA20ox2*			*sdw1.c*	[4]
4	19808	Denmark		Deba Abed			*HvGA20ox2*			*sdw1.c*	[4]
5	19691	Czechoslovakia		Diamant			*HvGA20ox2*			*sdw1.d*	[4]
6	30293	Australia		Franklin			*HvGA20ox2*			*sdw1.d*	[13]
7	31247	Great Britain		Tipple			*HvGA20ox2*			*sdw1.d*	[4]
8	31355	Germany		Braemar			*HvGA20ox2*			*sdw1.d*	[4]
9	20491	Great Britain		Golden Promise			*HvDep1*			*ari-e.GP*	[20]
10	26959	USA		Morex			wild type				[13]
11	19037	Sweden		Jotun			wild type				[6]
12	29576	USA		Bowman			wild type				[3]

## Data Availability

The data presented in this study are available in the figures and tables provided in the manuscript.

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
