# Peer review of "Determination of the Allelic Composition of the sdw1/denso (HvGA20ox2), uzu1 (HvBRI1) and ari-e (HvDep1) Genes in Spring Barley Accessions from the VIR Collection"

_plants, 2024, doi:10.3390/plants13030376_

Round 1
Reviewer 1 Report
Comments and Suggestions for Authors
Lodging of barley significantly limits its potential yield, and the dwarfness genes and loci are generally involved in lodging resistance such as genes sdw1/denso 11 (HvGA20ox2), uzu1 (HvBRI1), and ari-e (HvDep1) in barley. In the MS by Kseniia et al., the authors report on the design of 12 dCAPS markers for the sdw1.c and ari-e.GP alleles and the molecular screening of barley 13 accessions from the VIR collection. The results showed that the presence of dwarfing alleles in all genotypes determines high or medium 22 lodging resistance regardless of the influence of weather conditions based on these proposed dCASP markers. The MS is well written and organised, and I am happy to recommend an acceptance after a few of minor modifications with grammar improvements.
Comments on the Quality of English LanguageNo comments
Author Response
Thank you very much for taking the time to review this manuscript. We have added some information based on the your and another reviewer recommendations.
Figure S1 and Figure S2 in the supplementary material
page number - 6, paragraph - 2, and line 173
page number - 7, Figure 2
page number - 13, paragraph - 4, and line 364
page number - 12, paragraph - 1, and line 339
page number - 11, paragraph - 1, and line - 299
page number - 11, paragraph - 2, and line - 308

Reviewer 2 Report
Comments and Suggestions for Authors
(1) Table 1: draw two figures to show the height and logging resistance in 2021-2023 respectively.
(2) How the SNP/Indel were found is not clear. Which varieties were used in the gene sequence alignment? The author said the WT sequence was from Morex.
(3) All SNP/Indels should be identified from the barley pan-genome. The pan-genome has been released. The gene sequences including the promotor regions should be extracted from all the pan-genome varieties, and then make alignment to identify all the SNP/Indels.
(4) Identify the haplotypes of the genes based on the gene sequences or SNP/Indels.
(5) The markers should be developed to identify different haplotypes, not only alleles (need further analysis to identify the haplotypes).
(6) The relationships between the haplotypes and their phenotypes (plant height and lodging) are not clear. Any conclusions or suggestions which haplotypes combination is the best?
(7) KASP marker is more efficient and not expensive.
Author Response
Thank you very much for taking the time to review this manuscript. Please find the detailed responses below and the corresponding revisions in track changes in the re-submitted files.
Comments 1: Table 1: draw two figures to show the height and logging resistance in 2021-2023 respectively.
Response 1: Thank you for pointing this out. We have, accordingly, added 2 figures to emphasize this point. Both figures have been added to the supplementary material. Figure S1 and Figure S2.
Comments 2: How the SNP/Indel were found is not clear. Which varieties were used in the gene sequence alignment? The author said the WT sequence was from Morex.
Response 2: Indeed, the sequence of the wild-type allele of the HvGA20ox2 and HvDep1 genes was extracted from the whole genome assembly of the barley cv. Morex (MorexV3). To localize SNP/Indel for the sdw1.c and sdw1.d alleles, the data from the work by Y. Xu et al. (2017) were used. The whole genome assembly of the barley cv. Golden Promise (GPv1) we use as a source for ari-e.GP allele of HvDep1 gene. Additionally, we have made these clarifications in the text of the article: the names Figure 2 and 3, as well as in the materials and methods.
page number - 6, paragraph - 2, and line 173
page number - 7, Figure 2
page number - 13, paragraph - 4, and line 364
Comments 3: All SNP/Indels should be identified from the barley pan-genome. The pan-genome has been released. The gene sequences including the promotor regions should be extracted from all the pan-genome varieties, and then make alignment to identify all the SNP/Indels.
Response 3: Thanks for the comment, we have performed the alignment of the HvGA20ox2 gene for all 20 barley genotypes from pan-genome. As a result, we found three accessions (HOR8148, HOR10350 and HOR13821) with the same transposon insertion and single nucleotide substitution as the six accessions with the new sdw1.ins allele that we sequenced in this research. However, for these three samples from the pan-genome, the allelic composition of the HvGA20ox2 gene is not described. We have made an appropriate addition to the text of the article.
page number - 12, paragraph - 1, and line 339
Comments 4: Identify the haplotypes of the genes based on the gene sequences or SNP/Indels.
Response 4: The HvGA20ox2 (chromosome 3H), HvBRI1 (3H) and HvDep1 (5H) genes are located on a different chromosomes. Therefore, according to the definition of haplotype, (a group of alleles that are inherited together, according to Cox, C. B., Moore, P. D., & Ladle, R. J. (2016). p106. Biogeography: an ecological and evolutionary approach. John Wiley & Sons), we cannot distinguish haplotypes by alleles of these three genes. Instead of haplotypes in the studied genotypes, we give combinations of alleles of the genes HvBRI1, HvBRI1 and HvDep1 (see Table 1.)
Comments 5: The markers should be developed to identify different haplotypes, not only alleles (need further analysis to identify the haplotypes).
Response 5: The aim of this study was the provide the molecular screening of barley accessions from the VIR collection for identifying dwarfing alleles commonly used in breeding. To identify alleles of the sdw1/denso, uzu1 and ari-e genes, we use markers from literature sources. To adapt molecular screening to the terms to small practice-oriented laboratories, we have developed dCAPS markers for sdw1.c allele of the HvGA20ox2 gene and ari-e.GP of HvDep1 (instead of KASP markers). In this study, we did not set ourselves the task of studying the polymorphism of all gene regions, including promoters. This may be a task for future research.
Comments 6: The relationships between the haplotypes and their phenotypes (plant height and lodging) are not clear. Any conclusions or suggestions which haplotypes combination is the best?
Response 6: Thank you for pointing this out. We have added more information about this point. We cannot distinguish haplotypes by alleles of these three genes. Instead of haplotypes in the studied genotypes, we give combinations of alleles of the genes.
page number - 11, paragraph - 1, and line - 299
page number - 11, paragraph - 2, and line - 308
Comments 7: KASP marker is more efficient and not expensive.
Response 7: Indeed, a well-established KASP technique allows you to save time resources. However, the KASP analysis, in its turn, requires sophisticated equipment and expensive reagents, which are often unaffordable to small practice-oriented laboratories or are not officially supplied to a number of countries. We suggest using CAPS and dCAPS markers, as an alternative.

Round 2
Reviewer 2 Report
Comments and Suggestions for Authors
Comments:
(1) Describe how many alleles for each gene? , and which markers are used to differentiate these alleles?
(2)Delete the sentence "KASP is expensive".
Author Response
Thank you very much for taking the time to review this manuscript and correct our shortcomings. Please find the answers below and the corresponding changes in the change tracker in the resubmitted files. All corrections and additions to the text are highlighted in yellow.
Comments 1: Describe how many alleles for each gene?, and which markers are used to differentiate these alleles?
Response 1: Thank you for pointing this out. We have added some more information on this issue
page number - 2, paragraph - 2, and line 56, 76
page number - 3, paragraph - 2, and line 89, 93, 101
page number - 3, paragraph - 3, and line 109, 116, 121
Comments 2: Delete the sentence "KASP is expensive".
page number - 4, paragraph - 1, and line 125
